# CEREAL: FEW-SAMPLE CLUSTERING EVALUATION

## ABSTRACT

Evaluating clustering quality with reliable evaluation metrics like normalized mutual information (NMI) requires labeled data that can be expensive to annotate. We focus on the underexplored problem of estimating clustering quality with limited labels. We adapt existing approaches from the few-sample model evaluation literature to actively sub-sample, with a learned surrogate model, the most informative data points for annotation to estimate the evaluation metric. However, we find that their estimation can be biased and only relies on the labeled data. To that end, we introduce CEREAL, a comprehensive framework for few-sample clustering evaluation that extends active sampling approaches in three key ways. First, we propose novel NMI-based acquisition functions that account for the distinctive properties of clustering and uncertainties from a learned surrogate model. Next, we use ideas from semi-supervised learning and train the surrogate model with both the labeled and unlabeled data. Finally, we pseudo-label the unlabeled data with the surrogate model. We run experiments to estimate NMI in an active sampling pipeline on three datasets across vision and language. Our results show that CEREAL reduces the area under the absolute error curve by up to 78% compared to the best sampling baseline. We perform an extensive ablation study to show that our framework is agnostic to the choice of clustering algorithm and evaluation metric. We also extend CEREAL from clusterwise annotations to pairwise annotations. Overall, CEREAL can efficiently evaluate clustering with limited human annotations.

## 1 INTRODUCTION

Unsupervised clustering algorithms (Jain et al., 1999) partition a given dataset into meaningful groups such that similar data points belong to the same cluster. Obtaining high-quality clusterings plays an important role in numerous learning applications like intent induction (Perkins & Yang, 2019), anomaly detection (Liu et al., 2021), and self supervision (Caron et al., 2018). However, evaluating these clusterings can be challenging. Unsupervised evaluation metrics, such as Silhouette Index, often do not correlate well with downstream performance (von Luxburg et al., 2012). On the other hand, supervised evaluation metrics such as normalized mutual information (NMI) and adjusted Rand index (ARI) require a labeled reference clustering. This supervised evaluation step introduces a costly bottleneck which limits the applicability of clustering for exploratory data analysis. In this work, we study an underexplored area of research: estimating the clustering quality with limited annotations.

Existing work on this problem, adapted from few-sample model evaluation, can often perform worse than uniform random sampling (see Section 5). These works use learned *surrogate models* such as multilayer perceptrons to identify the most informative unlabeled data from the evaluation set. Similar to active learning, they then iteratively rank the next samples to be labeled according to an acquisition function and the surrogate model's predictions. However, many active sampling methods derive acquisition functions tailored to a specific classification or regression metric (Sawade et al., 2010; Kossen et al., 2021), which make them inapplicable to clustering. Furthermore, these methods only rely on labeled data to learn the surrogate model and ignore the vast amounts of unlabeled data.

In this paper, we present CEREAL (Cluster Evaluation with REstricted Availability of Labels), a comprehensive framework for few-sample clustering evaluation without any explicit assumptions on the evaluation metric or clustering algorithm (see Figure 1). We propose several improvements to the standard active sampling pipeline. First, we derive acquisition functions based on normalized mutual information, a popular evaluation metric for clustering. The choice of acquisition function depends on whether the clustering algorithm returns a cluster assignment (hard clustering) or a distribution over

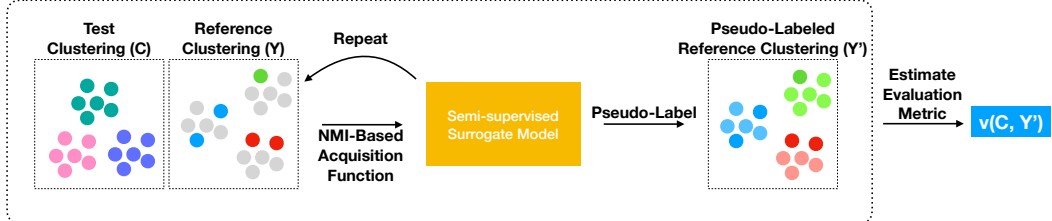

Figure 1: The CEREAL framework evaluates the test clustering with limited annotations for the reference clustering.

clusters (soft clustering). Then, we use a semi-supervised learning algorithm to train the surrogate model with both labeled and unlabeled data. Finally, we pseudo-label the unlabeled data with the learned surrogate model before estimating the evaluation metric.

Our experiments across multiple real-world datasets, clustering algorithms, and evaluation metrics show that CEREAL accurately and reliably estimates the clustering quality much better than several baselines. Our results show that CEREAL reduces the area under the absolute error curve (AEC) up to 78.8% compared to uniform sampling. In fact, CEREAL reduces the AEC up to 74.7% compared to the best performing active sampling method, which typically produces biased underestimates of NMI. In an extensive ablation study we observe that the combination of semi-supervised learning and pseudo-labeling is crucial for optimal performance as each component on its own might hurt performance (see Table 1). We also validate the robustness of our framework across multiple clustering algorithms – namely K-Means, spectral clustering, and BIRCH – to estimate a wide range evaluation metrics – namely normalized mutual information (NMI), adjusted mutual information (AMI), and adjusted rand index (ARI). Finally, we show that CEREAL can be extended from clusterwise annotations to pairwise annotations by using the surrogate model to pseudo-label the dataset. Our results with pairwise annotations show that pseudo-labeling can approximate the evaluation metric but requires significantly more annotations than clusterwise annotations to achieve similar estimates.

We summarize our contributions as follows:

- We introduce CEREAL, a framework for few-sample clustering evaluation. To the best of our knowledge, we are the first to investigate the problem of evaluating clustering with a limited labeling budget. Our solution uses a novel combination of active sampling and semi-supervised learning, including new NMI-based acquisition functions.

- Our experiments in the active sampling pipeline show that CEREAL almost always achieves the lowest AEC across language and vision datasets. We also show that our framework reliably estimates the quality of the clustering across different clustering algorithms, evaluation metrics, and annotation types.

## 2 RELATED WORK

**Cluster Evaluation** The trade-offs associated with different types of clustering evaluation are well-studied in the literature (Rousseeuw, 1987; Rosenberg & Hirschberg, 2007; Vinh et al., 2010; Gösgens et al., 2021). Clustering evaluation metrics - oftentimes referred to as validation indices - are either internal or external. Internal evaluation metrics gauge the quality of a clustering without supervision and instead rely on the geometric properties of the clusters. However, they might not be reliable as they do not account for the downstream task or make clustering specific assumptions (von Luxburg et al., 2012; Gösgens et al., 2021; Mishra et al., 2022). On the other hand, external evaluation metrics require supervision, oftentimes in the form of ground truth annotations. Commonly used external evaluation metrics are adjusted Rand index (Hubert & Arabie, 1985), V-measure (Rosenberg & Hirschberg, 2007), and mutual information (Cover & Thomas, 2006) along with its normalized and adjusted variants. We aim to estimate external evaluation metrics for a clustering with limited labels for the ground truth or the reference clustering. Recently, Mishra et al. (2022) proposed a framework to select the expected best clustering achievable given a hyperparameter tuning method and a computation budget. Our work complements theirs by choosing best clustering under a given *labeling* budget.

**Few-sample Evaluation**   Few-sample evaluation seeks to test model performance with limited access to human annotations (Hutchinson et al., 2022). These approaches rely on different sampling strategies such as stratified sampling (Kumar & Raj, 2018), importance sampling (Sawade et al., 2010; Poms et al., 2021), and active sampling (Kossen et al., 2021). Our work is closely related to few-sample model evaluation but focuses on clustering. Often, existing approaches tailor their estimation to the classifier evaluation metric of interest, which make them ill-suited to evaluate clustering (Sawade et al., 2010; Kossen et al., 2021). While we choose to focus on clustering, we do not make further assumptions about the evaluation metrics. Finally, a few approaches have used unlabeled examples to estimate performance of machine learning models (Welinder et al., 2013; Ji et al., 2021). These use a Bayesian approach along with unlabeled data to get a better estimate of precision and recall (Welinder et al., 2013), or group fairness (Ji et al., 2020) of learned models. In our work, we estimate a clustering evaluation metric and use the unlabeled data to train the surrogate model. Furthermore, we also propose novel acquisition functions that account for the distinctive properties of clustering and label them proportional to the underlying evaluation metric.

**Active Learning**   Active learning, a well-studied area of few-sample learning (Cohn et al., 1996; Settles, 2009; Gal et al., 2017; Kirsch et al., 2019; Ash et al., 2020; 2021), aims to train a model by incrementally labeling data points until the labeling budget is exhausted. Most of these methods differ in their acquisition functions to sub-sample and label data to train the models. We derive new clustering-specific acquisition functions that account for the distinctive properties of clustering (see Section 4). For a more in-depth introduction on active learning, we refer the reader to Settles (2009).

**Semi-supervised Learning**   Semi-supervised learning trains models using limited labeled data alongside large amounts of unlabeled data (Berthelot et al., 2019; Xie et al., 2020; Sohn et al., 2020). These methods use several strategies such as data augmentation and consistency regularization to effectively use the unlabeled data. In our work we use FixMatch (Sohn et al., 2020), a general-purpose semi-supervised learning method, and pseudo-labeling and show they are key to the success of our framework. We can also use a more domain-specific semi-supervised learning algorithm instead of FixMatch in CEREAL.

## 3   PRELIMINARIES

**Few-sample Cluster Evaluation**   Given a dataset, the problem of external evaluation metric estimates the clustering quality by how much a (learned) test clustering differs from another (annotated) reference clustering by comparing the cluster assignments. Assuming that a clustering is a function $f_c \colon \mathcal{X} \mapsto \mathcal{K}$ from a dataset $\mathcal{X} = \{x_1, x_2, ..., x_n\}$ to a set of clusters $\mathcal{K} = \{k_1, k_2, ..., k_m\}$, then we rewrite cluster evaluation as computing a scalar function $v$ between two clusterings: $v(f_c(\mathcal{X}), f_y(\mathcal{X}))$. In the few-sample case, one of the two clusterings, which we assume to be the reference clustering, is only partially known. For convenience, we denote:

$$C \coloneqq \{f_c(x) | \forall x : x \in \mathcal{X}\}, \qquad \text{the test clustering.}$$
$$Y \coloneqq \{f_y(x) | \forall x : x \in \mathcal{X}\}, \qquad \text{the reference clustering.}$$
$$Y_L \coloneqq \{f_y(x) | \forall x : x \in \mathcal{X}_L \subset \mathcal{X}\}, \quad \text{the partially labeled subset of the reference clustering.}$$
$$\mathcal{X}_U \coloneqq \mathcal{X} \setminus \mathcal{X}_L, \qquad \text{the subset of the dataset without reference labels.}$$

The problem then becomes estimating $v(C, Y)$ from the partial reference clustering, *i.e.* computing an estimate $\hat{v}(C, Y_L)$ such that $|v(C, Y) - \hat{v}(C, Y_L)|$ is minimized. While there are no formal requirements on $\mathcal{X}$ other than being clusterable, we assume for the rest of this paper that $\mathcal{X}$ is a set of vectors.

**Normalized Mutual Information**   Normalized mutual information (NMI) measures the mutual dependence of two random variables with a number in the interval $[0, 1]$. As outlined in Gösgens et al. (2021), it satisfies several desirable properties; it is easily interpretable, has linear complexity, is symmetric, and monotonically increases with increasing overlap to the reference clustering. In cluster evaluation, we treat the test clustering $C$ and the reference clustering $Y$ as random variables

---

**Algorithm 1:** Training algorithm for CEREAL.

---

**Input**  : Seed labeled data points $(\mathcal{X}_L, Y_L)$, unlabeled data points $\mathcal{X}_U$, number of queries $n$,
labeling budget $M$, acquisition function $\Phi$, surrogate model $\pi_{\boldsymbol{\theta}}$, test clustering $C$.
**Output** : $\hat{v}(C; Y)$.
$\pi_{\boldsymbol{\theta}} \leftarrow$ train the surrogate model with the seed labeling data
**while** $|Y_L| < M$ **do**

> $Y_L \leftarrow Y_L \cup \Phi(\pi_{\boldsymbol{\theta}}, f_c, \mathcal{X}_U, n)$; sample $n$ data points to label with the acquisition function
> $\pi_{\boldsymbol{\theta}} \leftarrow \mathcal{L}(\mathcal{X}_L, Y_L) + \mathcal{L}_U(\mathcal{X}_U)$; **reinitialize and train the semi-supervised surrogate model**
> $Y' \leftarrow Y_L \cup \pi_{\boldsymbol{\theta}}(\mathcal{X}_U)$; **pseudo-label the unlabeled data points**

**end**
**return** $\hat{v}(C, Y')$

---

and calculate the $\mathrm{NMI}(C, Y)$ as follows:

$$\mathrm{NMI}(C; Y) = \frac{\sum_{y \in Y} \sum_{c \in C} I_{CY}[c; y]}{(H_Y + H_C)/2}, \qquad H_Y = -\sum_{y \in Y} p(y) \log p(y),$$

$$I_{CY}[c; y] = p(c, y) \log \frac{p(c, y)}{p(y)p(c)}, \qquad H_C = -\sum_{c \in C} p(c) \log p(c).$$

Here, $p(c, y)$ is the empirical joint probability of the test cluster assignment being $c$ and the reference cluster assignment being $y$, $p(c)$ is the marginal of the cluster assignment, $p(y)$ is the marginal of the reference cluster assignment, and $H$ is the entropy. In our work, the goal is to estimate $\mathrm{NMI}(C, Y)$ from the labeled subset only, *i.e.* to compute an estimator $\widehat{\mathrm{NMI}}(C, Y_L)$ for the NMI of the full dataset.

## 4    THE CEREAL FRAMEWORK

In this section, we describe the CEREAL framework for few-sample clustering evaluation. The key idea is to modify a standard active sampling pipeline's surrogate function to incorporate concepts from semi-supervised learning.

**Surrogate Model**   Surrogate models in an active sampling pipeline are used to identify which informative unlabeled data points to label next. Typically, the surrogate model $\pi_{\boldsymbol{\theta}}$ is a $|\mathcal{K}|$-way classifier trained to minimize some loss function $\mathcal{L}(\mathcal{X}_L, Y_L)$ in order to approximate the reference model on $\mathcal{X}$. An acquisition function then uses the surrogate model to assign a score to each unlabeled data point. At each iteration a batch of $n$ data points is labeled based on these scores which grows the set $Y_L$. The process repeats until the labeling budget is reached. Finally, the metric is estimated with the final labeled data.

In this work, the surrogate model, trained with a semi-supervised learning algorithm, is used not only to acquire new labels but also pseudo-label the unlabeled data points. The surrogate model $\pi_{\boldsymbol{\theta}}$ is a $|\mathcal{K}|$-way classifier trained on both labeled *and* unlabeled data, *i.e.* $\mathcal{L}(\mathcal{X}_L, Y_L) + \mathcal{L}_U(\mathcal{X}_U)$. The use of unlabeled data in the surrogate model allows us to use any task-specific semi-supervised learning algorithm such as FixMatch (Sohn et al., 2020), MixMatch (Berthelot et al., 2019), or Noisy Student (Xie et al., 2020).

**Acquisition Functions**   Most acquisition functions aim to order candidate data points by their predictive uncertainty and/or expected information gain from labeling. Recently, Kossen et al. (2021) derived a cross entropy acquisition function for active testing, which accounts for uncertainties from both the surrogate model and the test model. However, cross entropy is suboptimal for clustering applications because at each iteration it assumes knowledge of the correct mapping between cluster labels from the surrogate model and test clustering. The total number of test clusters and reference clusters need not be the same in general, which exacerbates the problem.

We derive two novel acquisition functions based on NMI that account for the distinctive properties of clustering. NMI accounts for the label assignments of the test clustering and the surrogate model efficiently without searching for a direct mapping between them. Furthermore, in Section 5.1, we find that while estimating the NMI, sampling points proportional to the NMI can lead to greater reduction

in the error. Soft NMI. If the test clustering $f$ outputs a distribution over the cluster for a given data point $x_i$, then we use the Soft NMI acquisition function:

$$\Phi_{SNMI} := 1 - \sum_{y \in Y} \sum_{c \in C} f_c(c|x_i) \pi_{\boldsymbol{\theta}}(y|x_i) \frac{I_{CY}[c; y]}{(H_Y + H_C)/2}.$$

Hard NMI. If $f$ outputs a cluster assignment for a data point $x_i$, then the Soft NMI acquisition function reduces to the Hard NMI acquisition function:

$$\Phi_{HNMI} := 1 - \sum_{y \in Y} \pi_{\boldsymbol{\theta}}(y|x_i) \frac{I_{CY}[f_c(x_i); y]}{(H_Y + H_C)/2}.$$

**Pseudo-labeling**   The pseudo-labeling allows us to effectively use the unlabeled data in our evaluation metric estimation. We use the semi-supervised surrogate model to pseudo-label, or assign a hard label to the unlabeled data, i.e., $Y' = Y_L \cup \pi_{\boldsymbol{\theta}}(\mathcal{X}_U)$. Further, our pseudo-labeling approach does not require any additional thresholds commonly used in semi-supervised learning (Xie et al., 2020). Our experiments show that in many cases, this simple pseudo-labeling step reduces estimation error dramatically (see Section 5.1).

**Algorithm**   We describe CEREAL in an active sampling pipeline in Algorithm 1. Given an initial set of labeled data points to train the surrogate model, CEREAL iteratively uses the provided acquisition function to determine more samples to be labeled. Once the new samples are labeled, CEREAL retrains the surrogate model and uses the updated surrogate model to label the still unlabeled points. Finally, CEREAL estimates $v(C, Y)$ as outlined above.

## 5   EXPERIMENTS

In this section, we describe our experiments for few-sample cluster evaluation. First, we compare CEREAL to the active sampling baselines and show that we achieve better NMI estimates with fewer samples. Next, we show that our framework is robust to the choice of clustering algorithms and evaluation metrics. Finally, we extend CEREAL to pairwise annotation where we show that pseudo-labeling the data points with the surrogate allows us to estimate the NMI of a clustering. In our active sampling and robustness experiments, we use FixMatch (Sohn et al., 2020) as our semi-supervised algorithm (see Appendix A for more details).

### 5.1   ACTIVE SAMPLING EXPERIMENT

**Datasets**   We experiment with four datasets in vision and language, namely, MNIST (LeCun, 1998), CIFAR-10 (Krizhevsky et al., 2009), Newsgroup (Joachims, 1996), and arXiv arXiv.org submitters (2022). The MNIST dataset contains images of the digits from 0 to 9. The CIFAR-10 dataset contains images of animals and objects from 10 classes. The Newsgroup dataset contains texts of news articles from 20 classes. The arXiv dataset consists of 2.7 million abstracts with 20 top-level topics such as computer science, math, statistics, etc. For all datasets, we use the ground truth class labels as the reference clustering $Y$.

Before clustering the datasets, we compute a vector representation for each sample with off-the-shelf pretrained models. For MNIST and CIFAR-10, we pass the images through CLIP (Radford et al., 2021), a pretrained vision-language model, to get image representations. For Newsgroup and arXiv, we pass the text through SimCSE (Gao et al., 2021), a pretrained self-supervised model, to get text representations. Then, we reduce the dimensions to 128 with PCA using the FAISS library (Johnson et al., 2019). Finally, in this experiment, we cluster the representations with K-Means algorithm from scikit-learn (Buitinck et al., 2013). Since the arXiv dataset is large, we subsample 1 million data points before clustering with K-Means. We set $K = \{5, 10, 15\}$ for MNIST and CIFAR-10, $K = \{10, 20, 30\}$ for Newsgroup, and $K = \{15, 30, 50\}$ clusters.

**Baselines**   We compare CEREAL with strong active sampling baselines divided into two categories: active learning (MaxEntropy, BALD) and active testing baselines (CrossEntropy). The acquisition functions in the active learning baselines account only for the uncertainties or the information gain only from the surrogate model to produce the acquisition score. On the other hand, acquisition

functions in the active testing baselines account for the uncertainties from both the surrogate model and the test clustering to compute the acquisition score. We also consider Soft NMI and Hard NMI as active testing acquisition functions.

MaxEntropy (Shannon, 1948) computes the entropy of the unlabeled data point over clusters with the surrogate model. BALD (Houlsby et al., 2011) assigns higher acquisition scores to data points that maximize the information gain between the surrogate model's predictions and its posterior. For BALD, we use MCDropout (Gal & Ghahramani, 2016) to model uncertainty in the predictions. During the acquisition process, we run 10 inference cycles with the surrogate model and average the entropies over the inferences. The CrossEntropy acquisition function, introduced in Kossen et al. (2021), chooses points that maximize the cross entropy between the surrogate model and the test model. Here, we measure the cross entropy between the surrogate model and the test clustering $f_c(\mathcal{X})$. As suggested in Farquhar et al. (2021), we convert the scores from the acuqisition function to a distribution and then stochastically sample from the distribution to get an unbiased estimate of the evaluation metric.

**Model Architecture**    For these experiments, we use a 4-layer multilayer perceptron (MLP) with an input dimension and hidden dimension of 128 and a dropout of 0.2 after each layer for the surrogate model. Between each layer of the MLP, we apply a ReLU nonlinearity (Nair & Hinton, 2010). The classification head is a linear layer with output dimension equal to the number of clusters in the reference clustering.

**Training Details**    We use PyTorch to build our framework (Paszke et al., 2019). All our experiments are performed on a single Tesla T4 GPU with 16GB of memory. For fair comparison with active sampling baselines, we train our surrogate models in CEREAL in the same pipeline. First, we uniformly sample 50 data points, label them, and train the surrogate model on the resulting sample-label tuples. Then, we query 50 data points in each round with the acquisition functions until the labeling budget of 1000 data points is exhausted. We train the surrogate model on the training split with a cross entropy loss for 20 epochs and choose the best model based on the validation split. We optimize the model parameters using Adam (Kingma & Ba, 2015) with a batch size of $64$ and a learning rate of $0.01$ for all the datasets. We use the same pipeline and model architecture for FixMatch with a different set hyperparameters and optimization settings. For more training details, see Appendix A.

**Evaluation Criteria**    We want our NMI estimates to be good approximations of true NMI with few samples,*i.e.* we want to measure the error between our estimated NMI and the true NMI. To measure the error, inspired by the area under the learning curve (Cawley, 2011), we propose to compute the area under the absolute error curve (AEC) AEC as $\int_{|Y_L|}^{|Y_L|+M} |\mathrm{NMI}(C;Y) - \widehat{\mathrm{NMI}}(C;Y)|\, dl$ , where $Y_L$ is the seed labeled data and $M$ is the labeling budget in the active sampling pipeline. Since we acquire data points in a batch, we cannot simply add the absolute error and thus use the trapezoidal rule to compute the area under the curve between the two intervals.

**Results**    We summarize our results as follows: (1) CEREAL using NMI-based acquisition function and FixMatch almost always reports the lowest AEC compared to all the methods, (2) using only FixMatch or only pseudo-labeling in the framework can hurt performance, and (3) active sampling methods often perform worse than Random as they often overestimate or underestimate the NMI (see Table 1 and Figure 2).

NMI-based acquisition functions can help reduce the AEC compared to active-sampling baselines. As we see in Table 1, CEREAL with Soft NMI or Hard NMI achieves the lowest AEC compared to the existing active sampling baselines. CEREAL with Soft NMI reduces AEC up to 78.8% compared to uniform sampling and up to 74.7% compared to the best performing active sampling method, MaxEntropy. On CIFAR10, we observe that Soft NMI has the lowest AEC suggesting that our NMI based acquisition function can provide significantly accurate estimates.

FixMatch and pseudo-labeling, together, are the most critical components of CEREAL. In Table 1, we observe that only using FixMatch in an active sampling pipeline or only pseudo-labeling can hurt performance across datasets. In fact, when we compare Soft NMI, Soft NMI + FixMatch, and Soft NMI + Pseudo-labeling, we see that AEC *increases* on an average by $6.4\%$. But, Soft NMI

| Method | MNIST ↓ | CIFAR-10 ↓ | Newsgroup ↓ | arXiv ↓ |
|---|---|---|---|---|
| Random | $0.741_{0.07}$ | $0.549_{0.07}$ | $1.658_{0.13}$ | $2.133_{0.13}$ |
| BALD | $0.814_{0.08}$ | $0.723_{0.04}$ | $2.113_{0.14}$ | $2.183_{0.10}$ |
| MaxEntropy | $1.030_{0.03}$ | $1.574_{0.05}$ | $1.116_{0.08}$ | $1.544_{0.11}$ |
| CrossEntropy | $0.677_{0.05}$ | $2.055_{0.41}$ | $1.950_{0.07}$ | $1.230_{0.07}$ |
| Soft-NMI | $0.646_{0.06}$ | $\mathbf{0.348}_{0.03}$ | $1.625_{0.17}$ | $2.110_{0.14}$ |
| Hard-NMI | $0.759_{0.05}$ | $0.399_{0.05}$ | $1.522_{0.20}$ | $2.085_{0.12}$ |
| Random + FixMatch | $0.866_{0.07}$ | $0.504_{0.06}$ | $1.762_{0.12}$ | $2.075_{0.11}$ |
| Soft NMI + FixMatch | $0.701_{0.08}$ | $0.474_{0.06}$ | $1.646_{0.14}$ | $2.279_{0.10}$ |
| Hard NMI + FixMatch | $0.649_{0.05}$ | $0.431_{0.06}$ | $1.636_{0.14}$ | $1.978_{0.13}$ |
| Random + Pseudo-Labeling | $1.082_{0.11}$ | $0.885_{0.04}$ | $1.818_{0.06}$ | $1.147_{0.06}$ |
| Soft NMI + Pseudo-Labeling | $1.044_{0.12}$ | $0.820_{0.04}$ | $1.742_{0.09}$ | $1.354_{0.13}$ |
| Hard NMI + Pseudo-Labeling | $1.030_{0.13}$ | $0.790_{0.03}$ | $1.769_{0.12}$ | $1.296_{0.06}$ |
| CEREAL (Random + FixMatch + Pseudo-Labeling) | $\underline{0.331}_{0.02}$ | $\underline{0.393}_{0.03}$ | $\underline{0.729}_{0.03}$ | $0.490_{0.02}$ |
| CEREAL (Soft NMI + FixMatch + Pseudo-Labeling) | $0.345_{0.03}$ | $0.399_{0.03}$ | $\mathbf{0.707}_{0.03}$ | $\mathbf{0.453}_{0.02}$ |
| CEREAL (Hard NMI + FixMatch + Pseudo-Labeling) | $\mathbf{0.311}_{0.02}$ | $0.424_{0.02}$ | $0.734_{0.03}$ | $\underline{0.456}_{0.02}$ |

Table 1: Results for the benchmark evaluation comparing CEREAL with active sampling baselines. For each dataset, we report the average area under the absolute error curve (AEC) and the standard error across three clusterings and five random seeds. The method with the lowest AEC is in **bold** and the second lowest AEC is underlined.

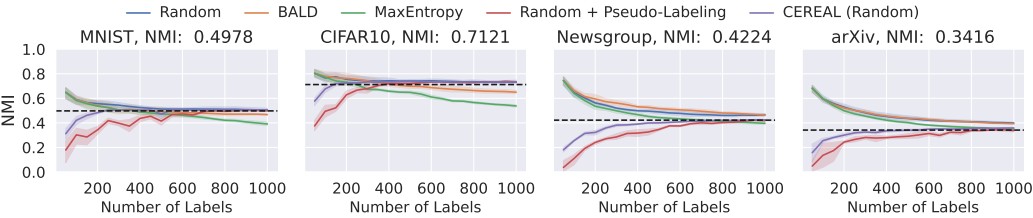

(a) Comparison of CEREAL (Random) using active learning baselines. We report the average NMI and the standard error across five random seeds.

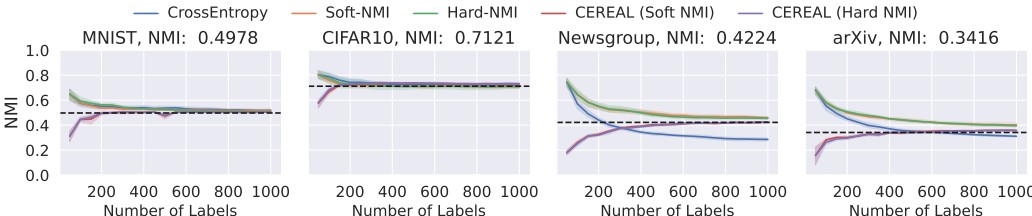

(b) Comparison of CEREAL using NMI-based acquisition functions with active testing baselines. We report the average NMI and the standard error across five random seeds.

Figure 2: Estimation curves as a function of labels.

with FixMatch and pseudo-labeling reduces the AEC by 46.6% on the MNIST dataset, 56.5% on the Newsgroup, and 78.5% on the arXiv dataset. These trends are similar across the Random and Hard NMI as well. While FixMatch is sufficient to improve performance of NMI acquisition functions + pseudo-labeling, we believe that any other appropriate semi-supervised learning algorithm would lead to similar improvements. Finally, we observe that the choice of acquisition function in CEREAL is less important than learning with FixMatch and pseudo-labeling the unlabeled dataset.

Existing active-sampling approaches often underestimate the NMI. In Figure 2a, we observe that both MaxEntropy and BALD can often underestimate the NMI which contributes to their AEC. We see that MaxEntropy on the Newsgroup dataset has a lower AEC compared to the other active sampling approaches but Figure 2a shows that it is incredibly biased and tends to underestimate the NMI. In Figure 2b, we observe that CrossEntropy on the Newsgroup dataset is severely biased and reports a lower NMI. As suggested earlier, CrossEntropy does not account for the mapping between the cluster assignments from the surrogate model and the test clustering. In contrast, we observe that CEREAL

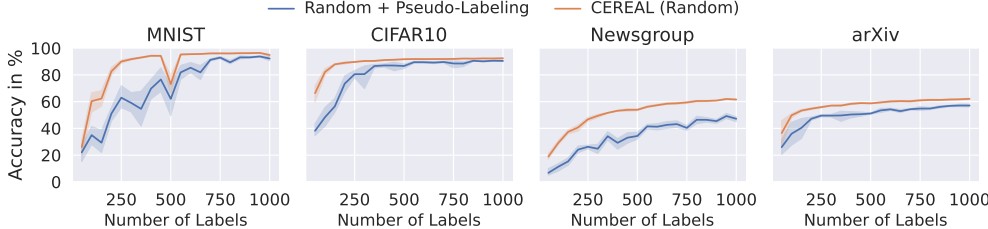

Figure 3: Accuracy of surrogate model

(Soft NMI) and CEREAL (Hard NMI) do not suffer from such biases. We also note that the Soft-NMI acquisition function overestimates NMI, which contributes to a lower AEC in the case of CIFAR-10.

**Accuracy of the Surrogate Models**    We investigate performance of the surrogate model in CEREAL used for pseudo-labeling. Figure 3 shows the accuracies of the surrogate model without and with FixMatch on the unlabeled data. We see that FixMatch is significantly better at pseudo-labeling the unlabeled data points. Comparing these with CEREAL in Figure 2a, we see a close relationship with performance of the surrogate model. This suggests that simply using a more sample efficient model can be a good way to pseudo-label clusters.

## 5.2    ROBUSTNESS OF CEREAL

We evaluate the robustness of CEREAL across clustering algorithms and evaluation metrics. Clustering algorithms differ in several ways including their preference to certain data types over others. For instance, K-Means can only cluster flat or spherical data whereas spectral clustering can also cluster non-flat structures. The same way, evaluation metrics including NMI can have biases which motivate data science practitioners to evaluate the clustering over several metrics (Gösgens et al., 2021).

**Setup**    We experiment with the Newsgroup dataset and choose several popular clustering algorithms and evaluation metrics. We consider the following clustering algorithms: K-Means, spectral clustering, and BIRCH. Spectral clustering uses the eigenvalues of the graph Laplacian to reduce the dimensions of the embeddings before clustering. This transductive clustering algorithm is helpful in clustering highly non-convex datasets. BIRCH is a type of hierarchical clustering algorithm typically useful for large datasets. For the evaluation metrics, we choose normalized mutual information (NMI), adjusted mutual information, and adjusted Rand index (ARI). ARI is a pairwise evaluation metric which computes the pairwise similarity between points, *i.e.* whether the test and reference clusterings agree on whether the points are in the same cluster or different clusters, normalized such that near-random test clusterings have scores close to zero. For simplicity, we compare Random and CEREAL (Random) by uniformly sampling and labeling 500 data points and then estimate the evaluation metric. We use the same model architecture as the previous section and choose $K = \{10, 20, 30\}$ for all the clustering algorithms. Then, we measure the absolute error between the estimated metric and the true evaluation metric with the complete dataset,*i.e.*, $|v(C, Y) - \hat{v}(C, Y)|$.

**Results**    Our results in Table 2 show that CEREAL consistently achieves lower error compared to Random. In Table 2a, we observe that CEREAL reduces the error up to 83% across different clustering algorithms. We also find that in Table 2b, CEREAL decreases the error up to 75% on

|  | Newsgroup | | |
| --- | --- | --- | --- |
| *Method* | *K-Means* ↓ | *Spectral* ↓ | *BIRCH* ↓ |
| Random | 3.36e-02 4.5e-03 | 3.71e-02 4.8e-03 | 3.12e-02 4.4e-03 |
| CEREAL (Random) | **9.42e-03** 7.7e-04 | **6.31e-03** 6.2e-04 | **2.88e-02** 2.0e-03 |

(a) Results comparing Random and CEREAL (Random) across multiple clustering algorithms. We report the absolute error between the true evaluation metric and estimated metric averaged across 3 clusterings for 3 evaluation metrics and 5 random seeds.

|  | Newsgroup | | |
| --- | --- | --- | --- |
| *Method* | *NMI* ↓ | *AMI* ↓ | *ARI* ↓ |
| Random | 7.17e-02 2.8e-03 | 1.97e-02 1.9e-03 | 1.04e-02 1.2e-03 |
| CEREAL (Random) | **1.79e-02** 2.2e-03 | **1.81e-02** 2.2e-03 | **8.53e-03** 7.8e-04 |

(b) Results comparing Random and CEREAL (Random) across multiple evaluation metrics. We report the absolute error between the true evaluation metric and estimated metric averaged across 3 clusterings from 3 clustering algorithms and 5 random seeds.

Table 2: Robustness of CEREAL across clustering algorithms and evaluation metrics.

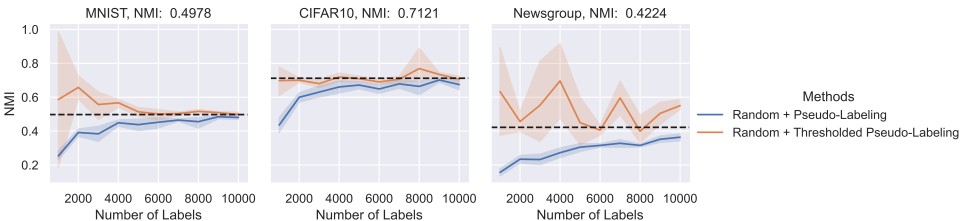

Figure 4: Estimation curves as a function of pairwise annotations.

different evaluation metrics. In particular, we see that CEREAL can approximate even ARI, a pairwise evaluation metric. Overall, our results show that CEREAL is robust across clustering algorithms and evaluation metrics.

### 5.3 EXTENSION TO PAIRWISE ANNOTATIONS

When the number of reference clusters is large, identifying the exact cluster for each data point can be a difficult and time consuming task. Additionally, the full set of ground truth clusters may not be known to annotators. This means that a human annotator needs to repeatedly decide if a data point belongs to an existing cluster or a new cluster. Therefore, we extend CEREAL from clusterwise annotations to pairwise annotations, *i.e.* whether or not two points belong to the same cluster.

Estimating clusterwise evaluation metrics such as NMI using only pairwise cluster information presents an additional challenge. We solve this practical issue by integrating CEREAL with L2C (Hsu et al., 2019) to learn a *multiclass* surrogate model from pairwise annotations in Algorithm 1. For more details on the L2C framework, see Appendix B. Even without exhaustively labeling all pairs of data points as in Hsu et al. (2019), we show that surrogate models learned with L2C are accurate enough to use in our few-sample evaluation problem.

**Setup**  As in previous sections we use MNIST, CIFAR10, and Newsgroup clusterings as our datasets. We uniformly sample 10,000 pairs of annotations in batches of size 1000. Since pairwise annotations for a dataset are sparse, a Random acquisition function samples about $(\mathcal{K} - 1)\times$ more negative than positive pairs. To improve generalization, during training we balance the samples to have an equal number of positive (same cluster) and negative (different cluster) pairs. In Figure 4, the $x$-axis denotes the number of annotated pairs *before* balancing the dataset. Here, the surrogate model is a linear classifier trained using Adam with a learning rate of $0.01$, weight decay of $5 \times 10^{-4}$, batch size of $32$ for 50 epochs. At each step, we train a surrogate model $\pi_{\boldsymbol{\theta}}$ and apply two pseudo-labeling strategies. Random + Pseudo-Labeling labels the entire dataset. Random + Thresholded Pseudo-Labeling only labels the data points with where $\max(\pi_{\boldsymbol{\theta}}(y|x)) > 0.5$.

**Results**  Figure 4 shows that our surrogate model with either pseudo-labeling strategy can be used to approximate the NMI with only pairwise annotations of the reference clustering. But, we see that thresholding with a fixed value can have higher variance in the estimates. Furthermore, our results also show that using weaker annotations comes at a cost. When we compare our results from Figure 2, we can see that CEREAL requires far fewer clusterwise labels to achieve similar NMI estimates.

## 6 CONCLUSION

In this paper, we introduced the problem of few-sample clustering evaluation. We derived CEREAL by introducing several improvements to the active sampling pipeline. These improvements consist of new acquisition functions, semi-supervised surrogate model, and pseudo-labeling unlabeled data to improve estimation of the evaluation metric.

Our work lays the foundation for future work for few-sample clustering evaluation. Interesting directions include: investigating the use of more powerful few-shot surrogate models in the active learning step (such as BERT (Devlin et al., 2018) for language and Flamingo (Alayrac et al., 2022) for vision) and developing more sophisticated sampling strategies for pairwise annotations to reduce the sample requirements further.

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

## A  FIXMATCH

We describe FixMatch (Sohn et al., 2020), the semi-superivsed learning algorithm used in the experiments in Section 5. This algorithm relies on two key components: consistency regularization and pseudo-labeling. The consistency regularization assumes that the model produces same output with different perturbations for an input data point. Pseudo-labeling assigns "hard" artificial labels to the unlabeled data with the trained model.

In our work, we train the surrogate model $\pi_{\boldsymbol{\theta}}$ with a cross entropy loss that leverages both labeled and unlabeled data. The overall loss function for a batch $B$ is

$$\mathcal{L}_{FM} = l_s + \lambda_u l_u \ ,$$

where $l_s$ is the loss on the labeled data, $l_u$ is the loss on the unlabeled data, and $\lambda_u$ is a scalar hyperparameter. The loss over the labeled dataset is defined as

$$l_s = \frac{1}{B} \sum_{i=b}^{B} H(y_b, \pi_{\boldsymbol{\theta}}(y|x_b)) \ ,$$

where $y_b$ is the reference cluster assignment, $\pi_{\boldsymbol{\theta}}(y|x_b)$ is the distribution of the cluster assignments for the data point $x_b$, and $H$ is the cross entropy between the reference cluster assignments and the predictions. The loss on the unlabeled dataset is

$$l_u = \frac{1}{\mu B} \sum_{b=1} \mathbb{1}(\max(q_b) \geq \tau) H(\hat{q_b}, \pi_{\boldsymbol{\theta}}(y|\mathcal{A}(x_b))) \ ,$$

where $\mu$ is a hyperparameter that controls the size of the unlabeled data relative to the labeled data, $\tau$ is a scalar threshold, $\max(q_b) = \max(\pi_{\boldsymbol{\theta}}(y|x_b))$ is the probability of the cluster assignment with the highest confidence with a weak augmentation like dropout, $\hat{q_b}$ is one hot vector with the peak on the highest cluster assignment, and $\mathcal{A}$ is a hard augmentation like mixup.

In this work, our input data points are embeddings rather than raw images or text data. This means the augmentations – weak and strong – have to be datatype agnostic. We consider Dropout (Srivastava et al., 2014) as our weak augmentation. Consistent with Appendix D.2 of Sohn et al. (2020), we use mixup (Zhang et al., 2018) as our strong augmentation in place of RandAugment/CTAugment, choose the mixup hyperparameter $\alpha = 9$, and mix random inputs only. While Sohn et al. (2020) validates the effectiveness of this "FixMatch + Input MixUp" method on CIFAR-10, mixup for text classification has been investigated in previous work (Guo et al., 2019; Sun et al., 2020; Yoon et al., 2021).

We set the following hyperparameters: $B = 32$, $\mu = 7$, $\tau = 0.95$, and $\lambda_u = 1$. We also use a cosine scheduler for the learning rate and set the warmup to 0 epochs. Finally, we train the model with the overall loss $\mathcal{L}_{FM}$ using SGD with a learning rate of $0.03$ and weight decay of $5 \times 10^{-4}$ for 1024 epochs.

### A.1  COMPUTATIONAL COST

Adding FixMatch to a small surrogate model (applied to the frozen input embeddings) is not expensive. Train time on a g4dn.xlarge (NVIDIA T4 16GB, $0.52 per hour) for a FixMatch model is 20 sec/100 samples (in our experiments, train time over $102\,450$ epochs scaled linearly with the number of samples; so it's 20 sec/100 samples, 40 sec/200 samples, etc.). While this dominates the overall compute cost of CEREAL (the other pipeline steps are in the sub 5sec range), it is still *extremely* cost efficient.

## B  L2C FRAMEWORK

We adapt the L2C framework to a realistic setting where we have limited pairwise annotations. The L2C framework (Hsu et al., 2019) uses only pairwise similarity annotations to train a multiclass classifier. They treat the multiclass classifier as a hidden variable and optimize the likelihood of the pairwise similarities.

Consider the set of pairwise similarities $S = \{\mathbb{1}(f_y(x_i) = f_y(x_j))|x_i \in \mathcal{X} \wedge x_j \in \mathcal{X}\}$ between pairs of $\mathcal{X}$. In a limited labeled setting, we have access only to $S_L$ where $S_L \subset S$. Suppose we do not

|                                                         | Random Initialization | Imbalanced Initialization | Balanced Initialization |
|---------------------------------------------------------|:---------------------:|:-------------------------:|:-----------------------:|
| Random                                                  | $1.658_{\ 0.13}$      | $1.899_{\ 0.12}$          | $1.922_{\ 0.17}$        |
| Soft-NMI                                                | $1.625_{\ 0.17}$      | $1.651_{\ 0.07}$          | $1.952_{\ 0.21}$        |
| Hard-NMI                                                | $1.522_{\ 0.20}$      | $1.740_{\ 0.12}$          | $2.131_{\ 0.27}$        |
| Random + FixMatch                                       | $1.762_{\ 0.12}$      | $1.660_{\ 0.09}$          | $1.867_{\ 0.20}$        |
| Soft NMI + FixMatch                                     | $1.646_{\ 0.14}$      | $1.539_{\ 0.15}$          | $1.868_{\ 0.17}$        |
| Hard NMI + FixMatch                                     | $1.636_{\ 0.14}$      | $1.595_{\ 0.11}$          | $1.951_{\ 0.12}$        |
| Random + Pseudo-Labeling                                | $1.818_{\ 0.06}$      | $1.919_{\ 0.19}$          | $1.923_{\ 0.07}$        |
| Soft NMI + Pseudo-Labeling                              | $1.742_{\ 0.09}$      | $2.166_{\ 0.11}$          | $2.024_{\ 0.19}$        |
| Hard NMI + Pseudo-Labeling                              | $1.769_{\ 0.12}$      | $2.096_{\ 0.09}$          | $1.948_{\ 0.26}$        |
| CEREAL (Random +FixMatch + Pseudo-Labeling)             | $0.729_{\ 0.03}$      | $1.090_{\ 0.04}$          | $\mathbf{0.999}_{\ 0.01}$ |
| CEREAL (Soft NMI +FixMatch + Pseudo-Labeling)           | $\mathbf{0.707}_{\ 0.03}$ | $\mathbf{1.077}_{\ 0.01}$ | $1.013_{\ 0.07}$    |
| CEREAL (Hard NMI +FixMatch + Pseudo-Labeling)           | $0.734_{\ 0.03}$      | $1.127_{\ 0.03}$          | $1.054_{\ 0.02}$        |

Table 3: Comparing the different strategies to acquire the initial labels in the active sampling pipeline. We report the average area under the absolute error curve (AEC) and the standard error across the clusterings on five random seeds. The method with loweest AEC is in **bold**.

know the number of clusters in the dataset, then we set the classification layer in $\pi_{\boldsymbol{\theta}}$ to a sufficiently large $\mathcal{K}$. When $s_{ij} = 1$ then $\pi_{\boldsymbol{\theta}}(x_i)$ and $\pi_{\boldsymbol{\theta}}(x_j)$ will have a sharp peak over the same output class. Therefore, we train the multiclass classifier $\pi_{\boldsymbol{\theta}}$ by optimizing the similarities with a binary cross entropy as follows:

$$\mathcal{L} = - \sum_{s_{ij} \in S_L} s_{ij} \log(\pi_{\boldsymbol{\theta}}(x_i)^T \pi_{\boldsymbol{\theta}}(x_j)) + (1 - s_{ij}) \log(1 - \pi_{\boldsymbol{\theta}}(x_i)).$$

Unlike the original formulation of the L2C framework, we have access to a limited number of pairwise annotations. We balance the dataset with an equal number of positive samples and negative samples. Finally, in Figure 4, we see that L2C with limited annotations can learn a multiclass classifier.

## C  COMPARISON TO HUMAN ANNOTATIONS

The cost of human annotations is significantly higher than the cost of GPU hours required to run CEREAL. For instance, consider the arXiv dataset with 2.7 million samples. Suppose we want to manually annotate 1 million uniform random samples at the cost of $0.05; this would cost $50,000. In practice, we often have multiple annotations for a sample, which would increase the cost between $100,000 and $150,000. Further, this would take an enormous amount of time for the annotators. On the other hand, our active sampling experiment for the arXiv dataset takes about 1 hour on a single GPU which costs about $0.52 per hour. Our algorithm often provides reliable estimates with around 500 samples which reduces the annotation cost to $25. Therefore, for a fraction of the cost, CEREAL can estimate the quality of the clustering reliably.

## D  LIMITATIONS

The CEREAL framework has a few limitations. Often, when the number of clusters is large, identifying the exact cluster for each data point can be difficult. While we use the pairwise extension of CEREAL to address this issue, it requires significantly more annotations to achieve similar NMI estimates.

We also see that while our method is robust across several clustering algorithms and evaluation metrics, the gains are nearly neutral for Adjusted Mutual Information (See Section E.3).

## E  ADDITIONAL EXPERIMENTS

### E.1  CLASS IMBALANCE

To understand the effect of label initialization, we conduct an ablation with different strategies without the surrogate model. We experiment with three label acquisitiion functions: random initialization, imbalanced initialization, and balanced initialization. In random initialization, we uniformly sample the data points and label them. In the imbalanced initialization, we randomly choose a clusters

| Method | MNIST ↓ | CIFAR-10 ↓ | Newsgroup ↓ | arXiv ↓ |
|---|---|---|---|---|
| Random | 300 | 200 | 650 | 1000 |
| BALD | 300 | 1000 | 1000 | 1000 |
| MaxEntropy | - | - | 400 | 550 |
| CrossEntropy | 250 | 1000 | - | 550 |
| Soft-NMI | 200 | **100** | 850 | 1000 |
| Hard-NMI | 300 | **100** | 500 | 1000 |
| Random + FixMatch | 350 | 150 | 800 | 1000 |
| Soft NMI + FixMatch | **150** | 200 | 700 | - |
| Hard NMI + FixMatch | 250 | 150 | 700 | 1000 |
| Random + Pseudo-Labeling | 550 | 250 | 600 | 600 |
| Soft NMI + Pseudo-Labeling | 500 | 250 | 600 | 700 |
| Hard NMI + Pseudo-Labeling | 550 | 250 | 650 | 650 |
| CEREAL (Random + FixMatch + Pseudo-Labeling) | **150** | **100** | **300** | 200 |
| CEREAL (Soft NMI + FixMatch + Pseudo-Labeling) | **150** | **100** | **300** | **150** |
| CEREAL (Hard NMI + FixMatch + Pseudo-Labeling) | **150** | **100** | **300** | 200 |

Table 4: Number of labeled samples required to reach $\pm 0.05$ of the true NMI value. A "-" indicates the method does not converge within this target accuracy. We report the median across five random seeds. The method with fewest number of samples is in **bold**.

and choose all the data points from the cluster till the seed labeling budget. Finally, in balanced initialization, we sample proportionally from all the clusters for the seed labels. In all three strategies, after the seed label initialization we follow the same Algorithm 1.

Our results in Table 3 show that random initialization, compared to other strategies, achieves the lowest AEC. We also observe the balanced sampling performs slightly better than the imbalanced sampling suggesting that a biased seed label set can hurt NMI estimation.

## E.2 Sample Complexity

Table 4 shows the number of labels necessary to converge within a target NMI $x$. We set $x = \pm 0.05$ of the true NMI value and calculate the median of the required data rounded to the nearest batch of 50 samples. CEREAL performs equal to or better than all baselines from Section 5.1.

## E.3 Robustness Across Clusterings

We evaluate the robustness of CEREAL on different evaluation metrics, clustering algorithms, and numbers of clusters on the Newsgroup dataset. We use the same setup as described in Section E.3.

**Evaluation Metrics** Figure 5 provides lower NMI and ARI estimates compared to the Random with limited labeled data. However, we observe that CEREAL might hurt performance on AMI estimation. We see this as a potential limitation of our work and requires further investigation (see AppendixD).

**Clustering Algorithms** We consider the same three clustering algorithms: K-means, Birch, and spectral clustering. Figure 6 shows that our method can reliably provide accurate NMI estimates compared to Random with as little as 200 to 250 labels on the dataset on all three types of clusterings.

**Number of Clusters** Figure 7 shows that CEREAL achieves lower error in NMI estimation compared to Random on two out of the three $k = \{10, 20, 30\}$. We see that when $k = 10$, CEREAL achieves a higher error compared to Random as it underestimates the performance. On the other hand, when the $k$ increases, we see that CEREAL provides precise NMI estimates. This result suggests that CEREAL could benefit when evaluating clusterings with a large number of clusters.

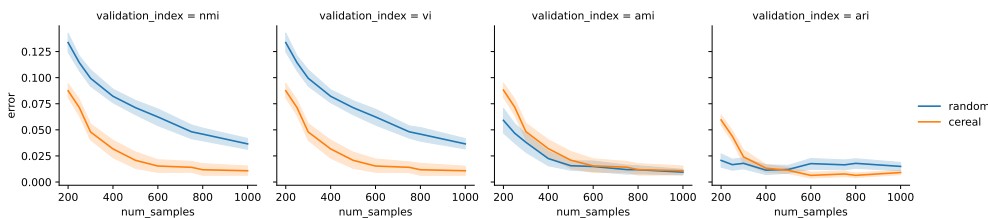

Figure 5: Epmstimation curves for varying clustering evaluation metrics.

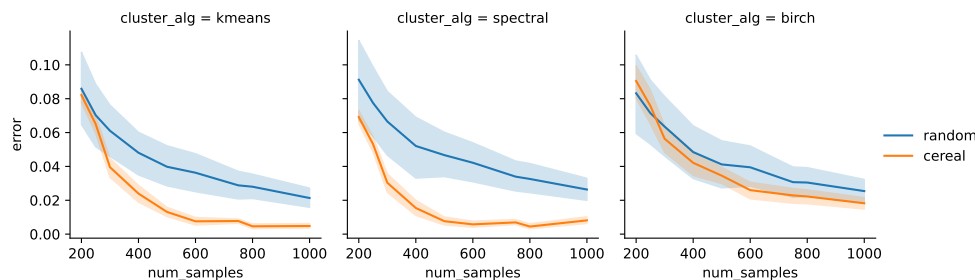

Figure 6: NMI estimation curves for varying clustering algorithms on the Newsgroups dataset.

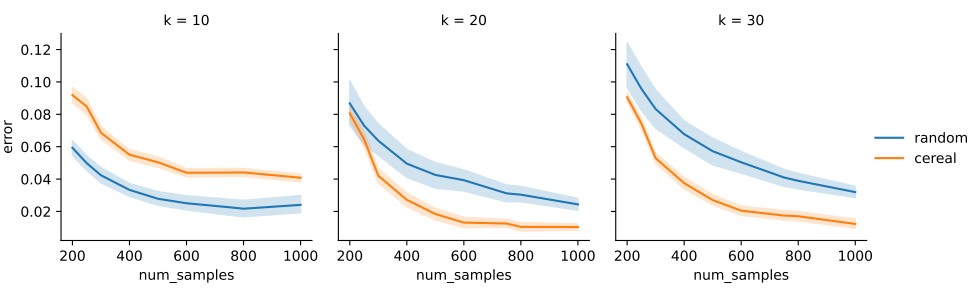

Figure 7: NMI estimation curves for varying number of clusters in the $K$-means test clustering on the Newsgroups dataset.

