# OpenReview forum: "CEREAL: Few-Sample Clustering Evaluation"
_ICLR.cc/2023/Conference — Submitted to ICLR 2023_

### Official Review · Reviewer_cxVs · 2022-10-21

**Confidence:** 3
**Correctness:** 3
**Technical Novelty And Significance:** 3
**Empirical Novelty And Significance:** 3
**Recommendation:** 6

**Clarity, Quality, Novelty And Reproducibility:**

# Clarity and Quality
* The paper is generally well-written and readable.

# Novelty
* To the best of my knowledge this is a novel approach to a novel problem.

# Reproducibility
* Not all of the important details for reproducibility are in the paper, but code is provided. However, some questions related to reproducibility of tuning procedures remain unanswered - see "Weaknesses" above.

**Strength And Weaknesses:**

# Strengths
* The paper is well-written and polished.
* To the best of my knowledge, the proposed method is novel and fills a gap in the literature (evaluating clustering methods with few labels).
* The ablation studies / comparisons in Table 1 are quite thorough.
* The proposed method seems to perform well in the scenarios where it has been tested.

# Weaknesses
* On page 2 there's talk of "experiments across multiple real-world datasets" but the paper uses MNIST, CIFAR-10, and Newsgroup, which I wouldn't consider "real-world" datasets. I found myself wondering why these datasets were used in the paper? Is the reason because these methods don't scale up well? If so that's fine, but it should be clearly identified as a limitation in the text.
* This is related to the previous point. As I reader, I had unanswered questions about the computational costs involved here. (1) How many GPU-hours does it take the run this algorithm from start to finish, and how does that scale with dataset size? (2) If we convert those GPU-hours to an amount of money, how does that cost compare the cost of just e.g. labeling the images on Mechanical Turk? The basic question is: "When is the algorithm worth it?" If the answer is "never, at least not with current methods" that's not a dealbreaker - it just needs to be spelled out clearly in the paper.
* If I understand correctly, "ground truth" is defined to be the k-means results using K={5, 10, 15} for MNIST/CIFAR-10 and K={10, 20, 30} for Newsgroup. Final performance numbers are averaged across these clusterings. How were these values of K chosen? They seem both arbitrary and very important for the final results.
* This AEC metric is a bit hard to interpret. Is this something being proposed by this paper? I didn't find references to it elsewhere. I think it would be better to compute error relative to the NMI value you're trying to estimate. i.e. are these estimates off by 1%? 10%? 50%? Hard to say from AEC. In addition, AEC also makes it hard to put the results in context because other papers seem to use different metrics.
* Is the learning rate not tuned? How was it selected? My concern is that by using the same learning rate for all methods, some might be getting "lucky" (because that's a good learning rate for them) while others might be getting "unlucky" (and would do better with a different learning rate). This would invalidate the claim that CEREAL is better than the competition.
* The "robustness" experiments were conducted on Newsgroup with 500 examples, which happens to be a setting where the proposed method does very well as we see in Figure 1(a). Is this a fair comparison? If not, please highlight in the text that this is cherrypicked.

# Minor Comments
* The text for the figures could be a little bigger.
* It would be nice if all of the "NMI vs. Number of Labels" figures all had the same y-axis range.
* I'm curious about the role of the initial embedding - I wonder how much things would change if we used e.g. ImageNet pretraining instead of CLIP.
* Since we're already using CLIP to embed the images, what if we pseudo-labeled the entire dataset with CLIP? Might that lead to a competitive baseline?
* Does $\Phi$ return the actual points selected, or the acquisition function values? There seems to be a bit of ambiguity about this between the text and the algorithm box.

**Summary Of The Paper:**

This paper considers the problem of evaluating the quality of a proposed clustering. The traditional approach (computing normalized mutual information, NMI) requires every sample to have a ground-truth cluster label. The goal of this paper is to approximate NMI with a limited number of ground-truth cluster labels. The proposed approach is to adapt ideas from active learning and semi-supervised learning to the clustering context. The paper performs experiments on 3 datasets (MNIST, CIFAR-10, Newsgroup) and compares a number of baselines against variants of the proposed approach.

**Summary Of The Review:**

This is a well-written, interesting paper that addresses an important gap in the literature. I enjoyed reading it and found it very informative as someone who doesn't do too much with clustering. Still, I have some questions about experimental methodology and significance. If these can be satisfactorily addressed I will be glad to see this paper accepted.

Dec 12, 2022 Update: I have revised my rating -- see comment below.

---

> ### Author Response · Authors · 2022-11-11
> **Response to Reviewer cxVs (1/2)**
>
> Thank you for your thoughtful feedback on our manuscript. We hope you will consider championing our work. We would like to address your questions in this review.
>
> > On page 2 there's talk of "experiments across multiple real-world datasets" but the paper uses MNIST, CIFAR-10, and Newsgroup, which I wouldn't consider "real-world" datasets. I found myself wondering why these datasets were used in the paper? Is the reason because these methods don't scale up well? If so that's fine, but it should be clearly identified as a limitation in the text.
>
> We have included an additional experiment with the arXiv dataset to show that our framework can scale up to large-scale datasets. The dataset has 2.7M samples with 20 classes. We uniformly sample 1M data points and get their embeddings from SimCSE and cluster them. Our results show that CEREAL outperforms the best-performing sampling method by 78% in AEC on the arXiv dataset.
>
>
> > This is related to the previous point. As I reader, I had unanswered questions about the computational costs involved here. (1) How many GPU-hours does it take the run this algorithm from start to finish, and how does that scale with dataset size? (2) If we convert those GPU-hours to an amount of money, how does that cost compare the cost of just e.g. labeling the images on Mechanical Turk? The basic question is: "When is the algorithm worth it?" If the answer is "never, at least not with current methods" that's not a dealbreaker - it just needs to be spelled out clearly in the paper.
>
> Thank you for the question. The cost of human annotations is significantly higher than the cost of GPU hours. For instance, consider the arXiv dataset with 2.7 million samples. For computational reasons, we uniformly sample 1 million samples. Suppose we want to manually annotate 1 million samples at the cost of \\$0.05, it would cost \\$50,000. In practice, we often have multiple annotations for a sample, which would increase the cost between \\$100,000 and \\$150,000. Further, this would take an enormous amount of time for the annotators. On the other hand, our algorithm often provides reliable estimates with around 500 samples which reduces the annotation cost to \\$25. Our active sampling experiment for the arXiv dataset takes about 1 hour on a single GPU which costs about \\$0.52 per hour. Therefore, for a fraction of the cost, CEREAL can estimate the quality of the clustering.
>
> > If I understand correctly, "ground truth" is defined to be the k-means results using K={5, 10, 15} for MNIST/CIFAR-10 and K={10, 20, 30} for Newsgroup. Final performance numbers are averaged across these clusterings. How were these values of K chosen? They seem both arbitrary and very important for the final results.
>
> For the MNIST, CIFAR-10, and Newsgroup, we choose the number of clusters +/- 50% of the true number clusters and the true clusters. This gives us  K={5, 10, 15} for MNIST and CIFAR-10 and K={10, 20, 30} for Newsgroup. Our results are similar across the different splits.
>
>
> > This AEC metric is a bit hard to interpret. Is this something being proposed by this paper? I didn't find references to it elsewhere. I think it would be better to compute error relative to the NMI value you're trying to estimate. i.e. are these estimates off by 1%? 10%? 50%? Hard to say from AEC. In addition, AEC also makes it hard to put the results in context because other papers seem to use different metrics.
>
> The AEC (area under the error curve) is inspired by Area under Learning Curve, which is common in active learning literature [a]. This metric evaluates not only the final error in NMI but also how quickly the error decreases with more labeled examples. Furthermore, we primarily compare our work with the active testing framework [b] that compares the full error curves of different methods without reducing them to a quantitative metric.
>
>
> > Is the learning rate not tuned? How was it selected? My concern is that by using the same learning rate for all methods, some might be getting "lucky" (because that's a good learning rate for them) while others might be getting "unlucky" (and would do better with a different learning rate). This would invalidate the claim that CEREAL is better than the competition.
>
> In our experiments we used the learning rates suggested by previous implementations of active learning baselines [c] and FixMatch [d]. Since few-sample clustering evaluation is an unsupervised setup, extensive hyperparameter tuning is less applicable than in a standard supervised learning setup with train/validation/test splits. Additionally, the function approximator and data are identical throughout all methods. Therefore we expect the choice of learning rate to have a similar impact across all methods, and it’s unlikely that one method will get luckier than the others.

---

> > ### Author Response · Authors · 2022-11-11
> > **Response to Reviewer cxVs (2/2)**
> >
> >
> > > The "robustness" experiments were conducted on Newsgroup with 500 examples, which happens to be a setting where the proposed method does very well as we see in Figure 1(a). Is this a fair comparison? If not, please highlight in the text that this is cherrypicked.
> >
> > You are correct. Our goal is to see if the performance of CEREAL deteriorates with the choice of clustering algorithms and evaluation metrics. Our results only confirm that with a modest amount of labeled samples, say 500, CEREAL is the preferred choice over Random.
> >
> >
> > > The text for the figures could be a little bigger…It would be nice if all of the "NMI vs. Number of Labels" figures all had the same y-axis range.
> >
> > Thank you for the suggestions. We have updated the text in the figures with bigger text and updated the figures with the same y-axis range.
> >
> >
> > > Since we're already using CLIP to embed the images, what if we pseudo-labeled the entire dataset with CLIP? Might that lead to a competitive baseline?
> >
> > CLIP requires the names of the clusters which might not be available. CLIP is a vision-language model that can be used for zero-shot inference. The model uses prompts in natural language with the names of the classes. However, this requirement is infeasible because the names of the clusters are not always known as apriori. If we knew the names of all the classes, we could potentially use CLIP for pseudo-labeling. This is a great future direction to explore.
> >
> >
> > > Does $\phi$  return the actual points selected, or the acquisition function values? There seems to be a bit of ambiguity about this between the text and the algorithm box.
> >
> > $\phi$ samples n data points to label rather than the acquisition scores. As discussed in the Experiments (see Baselines, Paragraph 3), the acquisition scores assigned to the unlabeled data points are converted to an acquisition distribution. Then, we stochastically sample n points to get an estimate of the evaluation metric.
> >
> > [a] Baseline methods for active learning. AISTATS 2011. http://proceedings.mlr.press/v16/cawley11a/cawley11a.pdf
> >
> > [b] Active Testing: Sample-Efficient Model Evaluation. ICML 2021. https://arxiv.org/abs/2103.05331
> >
> > [c] DeepAL: Deep Active Learning in Python. 2021 https://arxiv.org/abs/2111.15258
> >
> > [d] FixMatch-pytorch. https://github.com/kekmodel/FixMatch-pytorch

---

> > ### Comment · Reviewer_cxVs · 2022-11-16
> > **Response to author comments**
> >
> > Thank you for the responses! Some comments below.
> >
> > > We have included an additional experiment with the arXiv dataset to show that our framework can scale up to large-scale datasets. The dataset has 2.7M samples with 20 classes. We uniformly sample 1M data points and get their embeddings from SimCSE and cluster them. Our results show that CEREAL outperforms the best-performing sampling method by 78% in AEC on the arXiv dataset.
> >
> > > The cost of human annotations is significantly higher than the cost of GPU hours. For instance, consider the arXiv dataset with 2.7 million samples. For computational reasons, we uniformly sample 1 million samples. Suppose we want to manually annotate 1 million samples at the cost of \\$0.05, it would cost \\$50,000. In practice, we often have multiple annotations for a sample, which would increase the cost between \\$100,000 and \\$150,000. Further, this would take an enormous amount of time for the annotators. On the other hand, our algorithm often provides reliable estimates with around 500 samples which reduces the annotation cost to \\$25. Our active sampling experiment for the arXiv dataset takes about 1 hour on a single GPU which costs about \\$0.52 per hour. Therefore, for a fraction of the cost, CEREAL can estimate the quality of the clustering.
> >
> > These are nice additions. While we could quibble over the details of the cost estimate, I think the general point is clear enough. I would suggest that you add a section to the appendix that addresses the computational cost angle and explicitly identifies any related limitations in terms of how your approach scales with the number of classes, number of data points, etc. What practical limitations and advantages should users know about upfront?
> >
> > > For the MNIST, CIFAR-10, and Newsgroup, we choose the number of clusters +/- 50% of the true number clusters and the true clusters. This gives us K={5, 10, 15} for MNIST and CIFAR-10 and K={10, 20, 30} for Newsgroup. Our results are similar across the different splits.
> >
> > Thanks for the clarification. What happens if these values of K are off by a larger margin? Also, I'm noticing that all of the datasets have relatively small numbers of clusters - is that an important limitation somehow?
> >
> > > The AEC (area under the error curve) is inspired by Area under Learning Curve, which is common in active learning literature [a]. This metric evaluates not only the final error in NMI but also how quickly the error decreases with more labeled examples. Furthermore, we primarily compare our work with the active testing framework [b] that compares the full error curves of different methods without reducing them to a quantitative metric.
> >
> > Got it. I still think there are probably more readily interpretable metrics which might be worth including. For instance, as a reader I'm really interested in "the number of labels necessary to get within X% of the true NMI value" because it allows us to immediately see the practical advantages of one method over another. Why measure unintuitive surrogates like AEC when we can measure the true object of interest directly? These are just suggestions, but I don't find AEC to be especially appealing as a metric.
> >
> > > In our experiments we used the learning rates suggested by previous implementations of active learning baselines [c] and FixMatch [d]. Since few-sample clustering evaluation is an unsupervised setup, extensive hyperparameter tuning is less applicable than in a standard supervised learning setup with train/validation/test splits. Additionally, the function approximator and data are identical throughout all methods. Therefore we expect the choice of learning rate to have a similar impact across all methods, and it’s unlikely that one method will get luckier than the others.
> >
> > First, I'm not sure I agree that hyperparameters matter less in unsupervised learning - see e.g. the original SimCLR paper, which does extensive hyperparameter tuning and demonstrates that it is very important. Second, I don't see why it's reasonable for "the choice of learning rate to have a similar impact across all methods" - please explain further.
> >
> > > You are correct. Our goal is to see if the performance of CEREAL deteriorates with the choice of clustering algorithms and evaluation metrics. Our results only confirm that with a modest amount of labeled samples, say 500, CEREAL is the preferred choice over Random.
> >
> > Got it. Will you commit to highlighting the fact that it's a cherrypicked setting in the text? It's important that readers have that context, especially when we're talking about robustness claims.

---

> > > ### Author Response · Authors · 2022-11-19
> > > **Response to Reviewer cxVs' comments (1/2)**
> > >
> > > We appreciate your response to our reply. We would like to address your comments below:
> > >
> > > > These are nice additions. While we could quibble over the details of the cost estimate, I think the general point is clear enough. I would suggest that you add a section to the appendix that addresses the computational cost angle and explicitly identifies any related limitations in terms of how your approach scales with the number of classes, number of data points, etc. What practical limitations and advantages should users know about upfront?
> > >
> > > Thank you for your suggestion. We have updated our manuscript to describe the cost-benefit of using CEREAL over naive labeling (see Appendix C). This is one of the main advantages of our framework.
> > >
> > > We have updated our manuscript to include a limitations section (see Appendix D). Often, when the number of clusters is large, identifying the exact cluster for each data point can be difficult. While we use the pairwise extension of CEREAL to address this issue, it requires significantly more annotations to achieve similar NMI estimates. We have highlighted this limitation in the paper.
> > >
> > > > Thanks for the clarification. What happens if these values of K are off by a larger margin? Also, I'm noticing that all of the datasets have relatively small numbers of clusters - is that an important limitation somehow?
> > >
> > > Thanks for this great question. We have included the arXiv datasets when K=50 clusters when the true number of clusters is 20. Our results are again consistent with other results (see Table 1 and Figure 1).
> > >
> > > > Got it. I still think there are probably more readily interpretable metrics which might be worth including. For instance, as a reader I'm really interested in "the number of labels necessary to get within X% of the true NMI value" because it allows us to immediately see the practical advantages of one method over another. Why measure unintuitive surrogates like AEC when we can measure the true object of interest directly? These are just suggestions, but I don't find AEC to be especially appealing as a metric.
> > >
> > > As suggested, provide additional results for all the methods to be +/- 0.05 of the true NMI and calculate the median of the required data rounded to the nearest batch of 50 samples (see Appendix E.2).
> > >
> > > |                                             | MNIST | CIFAR10 | Newsgroup | arXiv |
> > > |---------------------------------------------|-------|---------|-----------|---------|
> > > | Random                                      | 300   | 200     | 650       | 1000  |
> > > | BALD                                        | 300   | 1000    | 1000      | 1000  |
> > > | MaxEntropy                                  | -     | -       | 400       | 550   |
> > > | CrossEntropy                                | 250   | 1000    | -         | 550   |
> > > | Soft-NMI                                    | 200   | 100     | 850       | 1000  |
> > > | Hard-NMI                                    | 300   | 100     | 500       | 1000  |
> > > | Random + FixMatch                           | 350   | 150     | 800       | 1000  |
> > > | Soft NMI + FixMatch                         | 150   | 200     | 700       | -     |
> > > | Hard NMI + FixMatch                         | 250   | 150     | 700       | 1000  |
> > > | Random + Pseudo-Labeling                    | 550   | 250     | 600       | 600   |
> > > | Soft NMI + Pseudo-Labeling                  | 500   | 250     | 600       | 700   |
> > > | Hard NMI + Pseudo-Labeling                  | 550   | 250     | 650       | 650   |
> > > | \sys(Random + FixMatch + Pseudo-Labeling)   | 150   | 100     | 300       | 200   |
> > > | \sys(Soft NMI + FixMatch + Pseudo-Labeling) | 150   | 100     | 300       | 150   |
> > > | \sys(Hard NMI + FixMatch + Pseudo-Labeling) | 150   | 100     | 300       | 200   |
> > >
> > > CEREAL performs equal to or better than all baselines from Table 1. Note that a “-” indicates the method does not converge within this target accuracy.
> > >
> > > > First, I'm not sure I agree that hyperparameters matter less in unsupervised learning - see e.g. the original SimCLR paper, which does extensive hyperparameter tuning and demonstrates that it is very important. Second, I don't see why it's reasonable for "the choice of learning rate to have a similar impact across all methods" - please explain further.
> > >
> > > We agree with you that varying the learning rate can improve the performance of the surrogate model. This means the surrogate model will provide better pseudo-labels and give good estimates of the NMI.

---

> > > > ### Author Response · Authors · 2022-11-19
> > > > **Response to Reviewer cxVs' comments (2/2)**
> > > >
> > > >
> > > > > Got it. Will you commit to highlighting the fact that it's a cherrypicked setting in the text? It's important that readers have that context, especially when we're talking about robustness claims.
> > > >
> > > > In Appendix E.3, we have included an additional section with a more detailed list of results to evaluate the robustness of our work. We evaluate CEREAL on different evaluation metrics, clustering algorithms, and even the number of clusters.  We include results with 200, 300, 400, 500, 600, 800, and 1000 labels.
> > > >
> > > > Our results show that, more often, CEREAL is better than Random across all the experiments. In the evaluation metric, CEREAL performs the best on NMI and VI metrics. However, we observe that CEREAL might hurt performance on AMI estimation. We have highlighted this as a limitation of CEREAL (see Appendix D). CEREAL is always better than Random across different clustering algorithms when estimating the NMI. Finally, when we vary the number of clusters, we see that CEREAL is better than Random when K=20, 30 but reports a higher error when K=10. This result suggests that CEREAL could benefit when evaluating clusterings with a large number of clusters.

---

> ### Comment · Reviewer_cxVs · 2022-12-12
> **Leaning towards acceptance**
>
> The paper may not be perfect or groundbreaking, but enough of my concerns have been resolved for me to tentatively recommend acceptance. The work seems to be competently executed, and I think there are people in our community would find it interesting.
>
> If other reviewers (who may know more than I do about this specific area) feel strongly that it should be rejected, I'm happy to hear those arguments as well.

---

### Official Review · Reviewer_W5ci · 2022-10-23

**Confidence:** 4
**Correctness:** 4
**Technical Novelty And Significance:** 1
**Empirical Novelty And Significance:** 2
**Recommendation:** 3

**Clarity, Quality, Novelty And Reproducibility:**

The explanation of the method is clear, but there is a lack of motivation for the proposed evaluation metric and the choice of experiments on toy datasets is poor.

**Strength And Weaknesses:**

The main strength of the paper is the simplicity of the approach. The proposed method (described in Algorithm 1) which consists in iteratively selecting labels to samples and assign pseudo-labels to unlabelled samples is simple to understand.

The paper contains lots of weaknesses. The first major weakness is the lack of motivation for the proposed method. In what real-world scenario would this evaluation approach be useful to the machine learning community? The evaluation approach might be interesting to the data mining or information retrieval communities but I do not know in what contexts.

The second major weakness is the way the experiments are performed. The chosen datasets are toy datasets (MNIST, CIFAR-10, Newsgroup datasets) and the models are pretrained (on ImageNet) or via self-supervised learning. It is not surprising to see pseudo-label approaches work well on these datasets since classes are easily separable.

**Summary Of The Paper:**

The paper proposes a method to evaluate the clustering quality when in semi-supervised contexts where the number of labeled examples is limited.

**Summary Of The Review:**

See the paragraph above.

---

> ### Author Response · Authors · 2022-11-11
> **Response to W5ci**
>
> Thank you for your thoughtful feedback on our manuscript. We would like to address your questions in this review.
>
> > The first major weakness is the lack of motivation for the proposed method. In what real-world scenario would this evaluation approach be useful to the machine learning community? The evaluation approach might be interesting to the data mining or information retrieval communities but I do not know in what contexts.
>
> Clustering is a key component in numerous areas of machine learning research such as intent induction[a], anomaly detection [b], and self supervision [c]. As mentioned in our introduction, evaluating these clusterings with supervised evaluation metrics such as NMI and ARI requires labeled reference clustering. Our work aims to address this problem.
>
>
> > The second major weakness is the way the experiments are performed. The chosen datasets are toy datasets (MNIST, CIFAR-10, Newsgroup datasets) and the models are pretrained (on ImageNet) or via self-supervised learning. It is not surprising to see pseudo-label approaches work well on these datasets since classes are easily separable.
>
> We agree with your explanation that pseudo-labeling approaches are a powerful baseline due to features from the pretrained models. However, CEREAL consistently beats the random + pseudo-labeling method in all experiments, including the more realistic arXiv dataset. Are there any additional feature representations and/or datasets that would lead to a more thorough experiment?
>
>
> [a] Dialog Intent Induction with Deep Multi-View Clustering. EMNLP 2019. https://arxiv.org/abs/1908.11487
>
> [b] Clustering With Outlier Removal. IEEE Transactions on Knowledge and Data Engineering 2021. https://arxiv.org/abs/1801.01899
>
> [c] Unsupervised Learning of Visual Features by Contrasting Cluster Assignments. NeurIPS 2020. https://arxiv.org/abs/2006.09882

---

### Official Review · Reviewer_6PbC · 2022-10-24

**Confidence:** 2
**Correctness:** 3
**Technical Novelty And Significance:** 3
**Empirical Novelty And Significance:** 2
**Recommendation:** 3

**Clarity, Quality, Novelty And Reproducibility:**

The paper is well written but the core methodology is unclear and very briefly described. \
The proposed method does not show much novelty.The proposed acquisition functions are small modifications from the previous work.The
The experiment seems to show that the proposed method is effective compared with other baselines.


**Strength And Weaknesses:**

Strength
1. This paper studies an interesting and important problem, which is the efficient evaluation for unsupervised methods.
2. The paper uses adequate experimental results to support the effectiveness of the method.
Weakness:
1.The novelty of the paper is limited. The two proposed acquisition functions are simple and straightforward extensions of NMI.
2. Section 4 which introduces the core methodology is very brief and it is not very clear how the proposed method addresses the clustering evaluation challenges.
3.The proposed approach is purely intuitive. It is hard to understand how and why it would work in general.


**Summary Of The Paper:**

This paper proposes an evaluation method to estimate the clustering quality with a small number of labeled samples. The samples are selected by two novel acquisition functions.

**Summary Of The Review:**

The paper studies a meaningful problem. But the proposed method lacks novelty, a thorough discussion of the core methodology design, and theoretical underpinning.

---

> ### Author Response · Authors · 2022-11-11
> **Response to Reviewer 6PbC**
>
> Thank you for your thoughtful feedback on our manuscript. We would like to address your questions in this review.
>
> > The novelty of the paper is limited. The two proposed acquisition functions are simple and straightforward extensions of NMI.
>
> Our approach uses NMI-based acquisition functions to overcome the limitations of previous work and to account for the distinctive properties of clustering. Our experimental results show that existing acquisition functions are biased and provide unreliable estimates of NMI, whereas our acquisition functions successfully reduce bias. While we believe acquisition function design is an important, novel contribution, this work is also the first to investigate the critical problem of evaluating clustering with a limited labeling budget.
>
> We would also like to point out that Reviewer Z3Vb has said that the “proposed framework and acquisition function with NMI ensures the work having certain novelty” and Reviewer 6PbC has said that our work is “a novel approach to a novel problem”.
>
>
> > Section 4 which introduces the core methodology is very brief and it is not very clear how the proposed method addresses the clustering evaluation challenges.
>
> We refer to Section 4 which discusses that “cross entropy is suboptimal for clustering applications because at each iteration it assumes the knowledge of the correct mapping between cluster labels from the surrogate model and the test clustering. Furthermore, the total number of test clusters and the reference clusters need not be the same in general which exacerbates the problem.” Our acquisition functions address all of these clustering evaluation challenges efficiently because they do not search for a mapping between test clusters and reference clusters. We have made this clearer in the updated version.
>
>
> > The proposed approach is purely intuitive. It is hard to understand how and why it would work in general.
>
> We argue that CEREAL is a general-purpose framework to evaluate clusterings. We would like to highlight that our results provide reliable estimates of clustering quality across vision and language datasets, as well as across different clustering algorithms and evaluation indices. While we focus on empirical comparison in this paper, theoretical analysis of acquisition functions for clustering evaluation is a very interesting direction for future work.

---

### Official Review · Reviewer_Z3Vb · 2022-11-03

**Confidence:** 4
**Correctness:** 4
**Technical Novelty And Significance:** 3
**Empirical Novelty And Significance:** 2
**Recommendation:** 5

**Clarity, Quality, Novelty And Reproducibility:**

The overall clarity and quality of the paper are good, well, and clearly written; experiments are clearly discussed.
The proposed framework and acquisition function with NMI ensures the work having certain novelty.

**Strength And Weaknesses:**

Strength:

1. The few-sample clustering evaluation is an interesting topic to explore, which also has practical importance.
2. The paper is well-written and easy to follow.
3. The proposed framework seems novel and achieves sound results. The acquisition function with NMI is newly proposed in the paper.

Weakness:

1. How to incorporate FixMatch in training the surrogate model is not clear. In FixMatch, there is a branch of strong augmentation on the input image x. How does this part work, specifically the strong-weak data augmentation branches, as the input of the surrogate model is a feature vector?

2. The initialization of labeled data is uniformly sampling 50 data points, which doesn't consider the class balance in ground-truth labels. The random sampling could largely be class imbalanced which would affect the training of the surrogate model and consequently affect the final results. It would be helpful if the evaluations were separated into a class-balanced setting and a class-imbalanced setting.

3. The effectiveness and robustness of the method are not fully evaluated. In table 1, Soft-NMI achieves the best result on CIFAR-10, better than utilizing FixMatch + Pseudo-labeling. Although the method achieves the best result on MNIST by quite a large margin.  on a more complicated CIFAR-10 dataset (same size, same number of classes), there is no advantage of utilizing FixMatch + Pseudo-labelling. Meanwhile, using FixMatch has the large risk of introducing wrong pseudo-labels for those unlabeled data and the training from scratch is always time-consuming, which would be emphasized as the dataset carries larger scope. This brings the question that: is it worth introducing the FixMatch+pseudo-labeling on larger evaluation datasets with more classes and complicated data distribution? Currently even on CIFAR-10, there is no guarantee that FixMatch or semi-supervised learning in this framework would benefit.

**Summary Of The Paper:**

This paper proposes a framework to do a few-sample clustering evaluation. The few-sample clustering evaluation refers to the case that only with a few labeled samples, how to do the evaluation on the clustering quality. The framework proposed follows three steps: 1. an acquisition function with NMI (normalized mutual information) is designed; 2. semi-supervised learning method like FixMatch is used to train the surrogate model, which utilizes labeled data and unlabeled data together 3. pseudo-labeling the unlabeled data with the learned surrogate model before estimating the evaluation metric.

**Summary Of The Review:**

My current decision on the paper is based on the weaknesses mentioned above, and it would be really helpful if the authors could provide further discussions.

---

> ### Author Response · Authors · 2022-11-11
> **Response to Reviewer Z3Vb**
>
> Thank you for your thoughtful feedback on our manuscript. We would like to address your questions in this review.
>
> > How to incorporate FixMatch in training the surrogate model is not clear. In FixMatch, there is a branch of strong augmentation on the input image x. How does this part work, specifically the strong-weak data augmentation branches, as the input of the surrogate model is a feature vector?
>
> We would like to point you to Appendix A (see the last paragraph). You are correct that the augmentations – weak and strong – have to be datatype agnostic. We consider Dropout as our weak augmentation and mixup [a] as our strong augmentation. Using mixup as strong augmentation is suggested in FixMatch (see Appendix D.2 in [b]). We will move these details to the main paper if space allows.
>
>
> > The initialization of labeled data is uniformly sampling 50 data points, which doesn't consider the class balance in ground-truth labels. The random sampling could largely be class imbalanced which would affect the training of the surrogate model and consequently affect the final results. It would be helpful if the evaluations were separated into a class-balanced setting and a class-imbalanced setting.
>
> Uniformly sampling initial data points is a standard practice in active learning experiments. The surrogate model trained with the initially labeled samples can then be used to non-uniformly sample from the pool of unlabeled data.
>
> > The effectiveness and robustness of the method are not fully evaluated. In table 1, Soft-NMI achieves the best result on CIFAR-10, better than utilizing FixMatch + Pseudo-labeling. Although the method achieves the best result on MNIST by quite a large margin. on a more complicated CIFAR-10 dataset (same size, same number of classes), there is no advantage of utilizing FixMatch + Pseudo-labelling. Meanwhile, using FixMatch has the large risk of introducing wrong pseudo-labels for those unlabeled data and the training from scratch is always time-consuming, which would be emphasized as the dataset carries larger scope. This brings the question that: is it worth introducing the FixMatch+pseudo-labeling on larger evaluation datasets with more classes and complicated data distribution? Currently even on CIFAR-10, there is no guarantee that FixMatch or semi-supervised learning in this framework would benefit.
>
> We would like to clarify these comments. In Table 1 and Figure 2b, we observe that Soft-NMI overestimates NMI, which contributes to a lower AEC in the case of CIFAR-10. Additionally, the Soft-NMI acquisition function is one of the main contributions of this work. Second, CEREAL is a general framework. We can think of FixMatch and pseudo-labeling as hyperparameters. Lastly, we have included an additional large-scale dataset: the arXiv dataset. The dataset has 2.7M samples with 20 classes. Our results show that CEREAL outperforms the best-performing sampling method by 78% in AEC on the arXiv dataset. We have updated the experiments section to include this dataset.
>
> [a] mixup: Beyond Empirical Risk Minimization. ICLR 2018. https://arxiv.org/abs/1710.09412
>
> [b] Fixmatch: Simplifying semi-supervised learning with consistency and confidence. NeurIPS 2020. https://arxiv.org/abs/2001.07685

---

> > ### Comment · Reviewer_Z3Vb · 2022-11-15
> > **Response to the Authors' Comments**
> >
> > Dear authors,
> >
> > Thanks for providing detailed discussions. I reviewed the comments and checked the revised paper, including the appendix. I summarized my feedback in the following:
> >
> > 1. I still hold concerns about the implementation details of FixMatch. FixMatch is original with images, which enables multiple choices of strong data augmentation like AutoAugment, and Cutout. Meanwhile, strong-weak data augmentation is the key to consistency regularization. However, FixMatch in this work is directly applied to the feature space. **My concerns are**: 1). Although the authors explain the strong data augmentation on the feature space refers to Mixup. It is still unclear how Mixup is applied and useful without knowing/mixing the ground-truth labels. 2). The feature space is expected to be invariant, at least with general variations shared by objects, especially with a pre-trained model. This leads to my concern that the invariance property on the feature space would invalidate the consistency regularization in FixMatch.
> >
> > 2. Following the above concerns, it comes to *the weakness 3.* I mentioned in the previous comments. I understand that authors provide a framework as CEREAL. However, providing a solid methodology for a framework is also important. **My original concern still holds** : the necessity and importance of adopting FixMatch (semi-supervised method to train the surrogate model) are not sufficiently evaluated. Only MNIST and CIFAR10 are evaluated on CV tasks (the added ArXiv dataset is not a CV task), which are a bit toy datasets, as also mentioned by other reviewers. Even on CIFAR10, the necessity of using FixMatch is not addressed (weaker performance compared with soft-NMI). I specifically address the evaluation of the necessity for FixMatch because the current training of FixMatch (or semi-supervised methods) is time-consuming and requires heavy computational resources. As the authors propose a new framework, it should be carefully evaluated whether it is worthy and valuable to incorporate such a heavy step in the framework.
> >
> > 3. For *weakness 2*, I suggest the authors consider developing a class-balanced setting and a class-imbalanced setting for evaluations in further work. The reasons are as provided in the previous comments.
> >
> > I will keep the original rating, which votes for rejection based on the aforementioned concerns.

---

> > > ### Author Response · Authors · 2022-11-19
> > > **Response to Reviewer Z3Vb's comments (1/2)**
> > >
> > > Thank you for your thoughtful response. We have addressed your concerns below.
> > >
> > > > Although the authors explain the strong data augmentation on the feature space refers to Mixup. It is still unclear how Mixup is applied and useful without knowing/mixing the ground-truth labels… This leads to my concern that the invariance property on the feature space would invalidate the consistency regularization in FixMatch.
> > >
> > > We appreciate your concerns about combining MixUp and FixMatch effectively. Thankfully the original FixMatch paper outlines how to extend FixMatch augmentation in a data-agnostic manner. Specifically, in Appendix D.2 (Datatype-Agnostic Data Augmentation), they propose to replace RandAugment/CTAugment strong augmentation with either mixup or Virtual Adversarial Training (VAT). They also show that FixMatch + Input MixUp has a classification error of $10.99 \pm 0.50$, which is superior to a MixMatch baseline of $11.05 \pm 0.86$.
> > >
> > > CEREAL follows exactly the same procedure: mix random _inputs only_ with $\alpha=9$. The only innovation is applying this extension to both image _and text_ datasets. MixUp for text classification has been investigated in previous works such as [https://arxiv.org/abs/1905.08941] [https://arxiv.org/abs/2106.08062]. We have updated the manuscript to include these details and references (see Appendix A).
> > >
> > > >  the necessity and importance of adopting FixMatch (a semi-supervised method to train the surrogate model) are not sufficiently evaluated.
> > >
> > > Comparing Rows 13-15 to Rows 10-12 of Table 1, it is clear that FixMatch is sufficient to improve performance over the combination of NMI acquisition functions + Pseudo-Labeling. Further, FixMatch can be thought of as a hyperparameter and replaced with any other appropriate semi-supervised learning algorithm.
> > >
> > > > Only MNIST and CIFAR10 are evaluated on CV tasks
> > >
> > > The arXiv dataset shows that the benefits of CEREAL extend to a large-scale, real-world dataset. Is there an additional CV dataset that would strengthen this claim?
> > >
> > > > I specifically address the evaluation of the necessity for FixMatch because the current training of FixMatch (or semi-supervised methods) is time-consuming and requires heavy computational resources. As the authors propose a new framework, it should be carefully evaluated whether it is worthy and valuable to incorporate such a heavy step in the framework.
> > >
> > > Adding FixMatch to a small surrogate model (applied to the frozen input embeddings) is not expensive. Train time on a g4dn.xlarge (NVIDIA T4 16GB, ~0.52USD/h) for a FixMatch model is 20sec/100 samples (in our experiments, train time over 1024 epochs scaled linearly with the number of samples; so it’s 20sec/100samples, 40sec/200samples, etc.). While this dominates the overall compute cost (the other pipeline steps are in the sub-5sec range), it is still _extremely_ cost-efficient.

---

> > > > ### Author Response · Authors · 2022-11-19
> > > > **Response to Reviewer Z3Vb's comments (2/2)**
> > > >
> > > > >  I suggest the authors consider developing a class-balanced setting and a class-imbalanced setting for evaluations in further work.
> > > > We do not expect class imbalance to be an issue in the initial surrogate model training (but will run anyway?)
> > > >
> > > > In Appendix E.1, we include a new experiment to understand the effect of label initialization and conduct an ablation with different strategies, namely, cluster balanced and cluster imbalanced. We experiment with three label acquisition functions: random initialization, imbalanced initialization, and balanced initialization. Our results (see below) show that random initialization, compared to other strategies, achieves the lowest AEC. We also observe the balanced sampling performs slightly better than the imbalanced sampling suggesting that a biased seed label set can hurt NMI estimation.
> > > >
> > > > |                                             | Random Initialization | Imbalanced Initialization | Balanced Initialization |
> > > > |---------------------------------------------|-----------------------|---------------------------|-------------------------|
> > > > | Random                                      | 1.658                 | 1.899                     | 1.922                   |
> > > > | Soft-NMI                                    | 1.625                 | 1.651                     | 1.952                   |
> > > > | Hard-NMI                                    | 1.522                 | 1.740                     | 2.131                   |
> > > > | Random + FixMatch                           | 1.762                 | 1.660                     | 1.867                   |
> > > > | Soft NMI + FixMatch                         | 1.646                 | 1.539                     | 1.868                   |
> > > > | Hard NMI + FixMatch                         | 1.636                 | 1.595                     | 1.951                   |
> > > > | Random + Pseudo-Labeling                    | 1.818                 | 1.919                     | 1.923                   |
> > > > | Soft NMI + Pseudo-Labeling                  | 1.742                 | 2.166                     | 2.024                   |
> > > > | Hard NMI + Pseudo-Labeling                  | 1.769                 | 2.096                     | 1.948                   |
> > > > | CEREAL (Random + FixMatch + Pseudo-Labeling)   | 0.729                 | 1.090                     | 0.999                   |
> > > > | CEREAL (Soft NMI + FixMatch + Pseudo-Labeling) | 0.707                 | 1.077                     | 1.013                   |
> > > > | CEREAL (Hard NMI + FixMatch + Pseudo-Labeling) | 0.734                 | 1.127                     | 1.054                   |

---

### Author Response · Authors · 2022-11-11
**To all reviewers**

Thank you to all the reviewers for your thoughtful comments on our work. Here, we summarize the main clarifications and the major changes in the manuscript. Please see individual replies for more details.

### Additional Large-Scale dataset
Reviewers 6Pbc and cxVs have raised concerns about the choice of datasets used in the paper. To address this issue, we ran experiments with an additional large-scale dataset: the arXiv dataset. The dataset has 1M samples with 20 classes. Our results show that CEREAL outperforms the best-performing sampling method by 78% in AEC on the arXiv dataset. This is much better than the previously reported improvement of 57%. We have updated the manuscript to include this dataset.

### Motivation and Novelty
Reviewer W5ci has asked about the motivation of our work, i.e., a framework for few-sample clustering evaluation. We would like to point out that the problem is not only an underexplored area of research but also has practical uses in intent induction, anomaly detection, and self supervision. To the best of our knowledge, we are the first to address the importance of clustering quality estimation with limited labels. We also note that Reviewer Z3Vb has said that our work is an “interesting topic to explore, which also has practical importance.”, Reviewer 6PbC has said that our work studies an “interesting and important problem”, and Reviewer cxVs has said that our work “fills a gap in the literature”.

Reviewer 6PbC has questioned the novelty of the paper. We overcome the limitations of existing acquisition functions and propose novel NMI-based acquisition functions that account for the distinctive properties of clustering. Our results show that existing acquisition functions are biased and provide unreliable estimates of different evaluation metrics whereas our acquisition functions are less biased. We would also like to highlight that Reviewer cxVs has said that our work is “a novel approach to a novel problem” and Reviewer Z3Vb has said that the “proposed framework and acquisition function with NMI ensures the work having certain novelty”.

---

### Decision · Program_Chairs · 2023-01-20

**Decision:**

Reject

**Justification For Why Not Higher Score:**

First of all, I would like to apologize the lack of discussion among 2 reviewers and authors because the author provided an extensive author feedback. Most of the concerns have been addressed in the rebuttal phase, but the reviewers are not fully convinced with the answers. Moreover, the paper has been changed significantly; hence the paper should be reviewed again before being accepted.

**Justification For Why Not Lower Score:**

N/A

**Metareview: Summary, Strengths And Weaknesses:**

# Summary
This paper considers the problem of evaluating the quality of a proposed clustering. The traditional approach (computing normalized mutual information, NMI) requires every sample to have a ground-truth cluster label. The goal of this paper is to approximate NMI with a limited number of ground-truth cluster labels. The proposed approach is to adapt ideas from active learning and semi-supervised learning to the clustering context. The paper performs experiments on 3 datasets (MNIST, CIFAR-10, Newsgroup) and compares a number of baselines against variants of the proposed approach.
# Strengths:
- The few-sample clustering evaluation is an interesting topic to explore, which also has practical importance.
- The paper is well-written and easy to follow.
-The acquisition function with NMI is newly proposed in the paper.
- To the best of my knowledge, the proposed method is novel and fills a gap in the literature (evaluating clustering methods with few labels).
- The proposed method seems to perform well in the scenarios where it has been tested.
# Weaknesses:
1.	How to incorporate FixMatch in training the surrogate model is not clear. In FixMatch, there is a branch of strong augmentation on the input image x. How does this part work, specifically the strong-weak data augmentation branches, as the input of the surrogate model is a feature vector?
2.	The initialization of labeled data is uniformly sampling 50 data points, which doesn't consider the class balance in ground-truth labels. The random sampling could largely be class imbalanced which would affect the training of the surrogate model and consequently affect the final results. It would be helpful if the evaluations were separated into a class-balanced setting and a class-imbalanced setting.
3.	The effectiveness and robustness of the method are not fully evaluated. In table 1, Soft-NMI achieves the best result on CIFAR-10, better than utilizing FixMatch + Pseudo-labeling. Although the method achieves the best result on MNIST by quite a large margin. on a more complicated CIFAR-10 dataset (same size, same number of classes), there is no advantage of utilizing FixMatch + Pseudo-labelling. Meanwhile, using FixMatch has the large risk of introducing wrong pseudo-labels for those unlabeled data and the training from scratch is always time-consuming, which would be emphasized as the dataset carries larger scope. This brings the question that: is it worth introducing the FixMatch+pseudo-labeling on larger evaluation datasets with more classes and complicated data distribution? Currently even on CIFAR-10, there is no guarantee that FixMatch or semi-supervised learning in this framework would benefit.
4.	On page 2 there's talk of "experiments across multiple real-world datasets" but the paper uses MNIST, CIFAR-10, and Newsgroup, which I wouldn't consider "real-world" datasets. Is the reason because these methods don't scale up well? If so that's fine, but it should be clearly identified as a limitation in the text.
5.	This AEC metric is a bit hard to interpret. Is this something being proposed by this paper? I didn't find references to it elsewhere. I think it would be better to compute error relative to the NMI value you're trying to estimate. i.e. are these estimates off by 1%? 10%? 50%? Hard to say from AEC. In addition, AEC also makes it hard to put the results in context because other papers seem to use different metrics.
6.	Is the learning rate not tuned? How was it selected? My concern is that by using the same learning rate for all methods, some might be getting "lucky" (because that's a good learning rate for them) while others might be getting "unlucky" (and would do better with a different learning rate). This would invalidate the claim that CEREAL is better than the competition.
7.	The "robustness" experiments were conducted on Newsgroup with 500 examples, which happens to be a setting where the proposed method does very well as we see in Figure 1(a). Is this a fair comparison? If not, please highlight in the text that this is cherrypicked.